# A Method for Determining the Directional Angle of a Railway Route Based on Field Measurements

**DOI:** 10.3390/s24041131

**Published:** 2024-02-09

**Authors:** Wladyslaw Koc

**Affiliations:** Faculty of Civil and Environmental Engineering, Gdansk University of Technology, ul. G. Narutowicza 11/12, 80-233 Gdansk, Poland; kocwl@pg.edu.pl

**Keywords:** railway track, directional angle of the route, calculation method, issue of the length of the moving chord, assessment of the accuracy of the proposed method

## Abstract

The most effective method for determining the coordinates of the railway track axis is based on using mobile satellite measurements. However, there are situations in which the satellite signal may be disturbed (due to field obstructions) or completely disappear (e.g., in tunnels). In these situations, the ability to measure the value of the directional angle of a moving rail vehicle using an inertial system is useful. The directional angle is determined on a topographic map as the angle between the direction of the vehicle’s longitudinal axis (or the direction of a tangent to the track axis) and the reference direction, which is the north. This article presents a method for determining the directional angle of a railway line based on appropriate measurement data. The latter should be Cartesian coordinates of the track axis, allowing for the visualization of a given railway route and permitting a general orientation of its course to be obtained. The presented proposal for solving the problem refers to the assumptions made in the method for determining the curvature of the railway track axis using the moving chord. The assumptions of the proposed method for determining the directional angle of the railway route are discussed, along with the appropriate computational algorithms. The accuracy of this method is assessed using the adopted model geometric layout. Reference is also made to the appropriate method for determining the curvature of the railway track axis. In conclusion, we provide an example of determining the directional angle based on measurement data.

## 1. Introduction

During many years of operation, the shape of existing railway lines gradually changes and, after some time, begins to differ from its original layout, designed in accordance with applicable regulations. In terms of passenger comfort and operational safety, deviations of the track axis in the horizontal plane may be particularly unfavorable. Therefore, it is necessary to determine the current geometric shape of the route so that any irregularities can be eliminated and any corrections made. Appropriate measurement systems and computational algorithms serve this purpose.

The measurement methods currently used are similar in various railway administrations [1,2,3,4,5]. In addition to classic geodetic techniques, stationary satellite measurements are used based on the global navigation satellite system (GNSS) technique, which uses the so-called active geodetic networks [6]. Mobile satellite measurement methods are also being introduced [7], in which, in addition to GNSS receivers, inertial navigation system (INS) devices [8] and vision methods such as terrestial laser scanning (TLS) [9] are often used as supporting devices. Research is underway on the possibility of using systems composed of satellite receivers mounted on various types of vehicles [10,11].

Based on the measurements performed, it is now possible to identify a given route. This involves determining its geometric parameters by using the designated Cartesian coordinates of the track axis points in the appropriate national spatial coordinate system. For example, in Poland—in relation to plane coordinates—the PL-2000 system [12] is used. This system was created on the basis of assigning points on the GRS 80 (a geodetic reference system) reference ellipsoid [13] to appropriate points on the plane according to the Gauss-Krüger projection theory [14]. From a mathematical point of view, this assignment is unambiguous.

There is no doubt that in the process of identifying a given railway line, determining the track axis coordinates is the most important (and most time-consuming) element. However, from a practical point of view, knowing the coordinates alone does not provide much. At most it enables the visualization of a given route, determining its general location. Full identification of the measured route requires the use of appropriate computational algorithms. When considering a horizontal plane, the basis of analysis is the determined values of the eastern coordinates *Y_i_* and northern coordinates *X_i_* of a given measurement point on the plane in the PL-2000 system.

The current approach to the problem of identifying a measured railway route is to generate an optimized geometric layout in a given area by minimizing the deviations of this layout from the measurement points (while meeting the appropriate maintenance and operational requirements). The traditional solution to a design problem relies heavily on human experience [15,16]. Many works attempt to automate this process. There are two variants of this procedure: geometric identification [17,18,19,20] and alignment optimization [21,22,23,24,25,26]. In the works [27,28,29], several methods were proposed to combine these variants into an iterative process of reconstructing the track axis.

As it turns out, the problem of geometric identification of a railway track can be approached in a completely different way, moving away from matching the hypothetical model layout to the track axis points determined by direct measurements. The basis for identification may be the curvature of the geometric layout; thus, an appropriate method for determining the curvature should be developed. In [30], a proposal for a new method for determining the curvature of the track axis, named “the moving chord method”, was presented.

However, in each case, the identification of the track’s geometric layout is based on Cartesian coordinates determined in field measurements. Most often, these measurements are obtained using a ground-based measurement network, although GNSS measurements carried out in a static or mobile manner are becoming more and more popular. It seems that the most effective method for determining the coordinates of a railway track axis is through the use of mobile satellite measuremnts. This method is definitely the fastest and least labor-intensive, and it provides an incomparable amount of measurement data in relation to other methods. The lesser popularity of this method is probably due to the fact that its accuracy has not been fully clarified.

Meanwhile, in [7] this was largely explained. This work, using reliability theory, presented an analysis of statistical errors when positioning a railway route using mobile satellite measurements. The values of availability, reliability and continuity of GNSS positioning obtained in the conducted measurement campaigns were compared, from the point of view of the required accuracy in relation to three levels of precision: 1 cm (for track deformation diagnostics), 3 cm (for design purposes) and 10 cm (for railway line inventory). The thesis was confirmed that, with careful planning of the measurement campaign (ensuring an appropriate constellation of satellites during measurements), it is possible to achieve very high positioning accuracy. This mainly refers to the horizontal plane (i.e., 2D measurements), whose Cartesian coordinates constitute the basis for identifying the geometric layout. All positioning measures for the 3 cm and 10 cm levels were found to be close to 100%. For the 1 cm accuracy level, there is also very high availability, especially for the 2D level.

However, there are situations when the satellite signal may be disturbed (due to field obstructions) or completely disappear (e.g., in tunnels). Then the ability to measure the value of the directional angle Φ of the route using the inertial system is useful. Knowing this angle and carrying out the simultaneous measurement of the distance Δ*L* allows one to determine the Cartesian coordinates of the next measurement point (if one knows the coordinates of the previous point). The following formulas apply to the PL-2000 system:(1)$$Y_{i+1}=Y_{i}+\Delta L\cdot \cos\Phi_{i÷i+1}$$
(2)$$X_{i+1}=X_{i}+\Delta L\cdot \sin\Phi_{i÷i+1}$$

If a point lies on a curve, calculating its coordinates using Equations (1) and (2) may result in some inaccuracies. However, these inaccuracies cannot be particularly important; because the inertial system determines the values of the directional angle with a very high frequency, hence the distances Δ*L* to the previous point are relatively small. Unlike maritime navigation, the issue of height differences is not a problem because the longitudinal inclinations of tracks on railway lines are very gentle (expressed per mile). A separate problem may be the accuracy of the inertial system, which may be limited in certain unfavorable conditions (e.g., in deep tunnels).

This article aims to present the principles of using the directional angle in the process of geometric identification of a railway track axis.

## 2. Directional Angle of the Railway Route

The concept of a directional angle is particularly relevant in the field of traditional seafaring. In this field, the term ”diametrical line” is used. This imaginary line connects the bow and stern of the vessel, located in the ship’s symmetry plane. The diametrical line has no specific height in this plane, but is parallel to the horizon. This is directly related to the concept of course, i.e., the direction (horizontal) in which the ship is steered or should be steered. The heading is determined as a measure of the angle (in degrees, clockwise) from the north reference direction. Depending on the reference direction, the following headings may be measured: true heading, magnetic heading, compass heading and gyrocompass heading. True course (TC) is the angle between true north (geographic meridian) and the front of the ship’s midline.

It is easy to see that the concept of a diametrical line can also be applied to the longitudinal axis of a railway wagon, defined by the line connecting the turning pins of bogies or wheel sets. The angle of inclination of the diametrical line relative to the north can therefore be treated as the directional angle (i.e., heading) of the rail vehicle. On straight sections of the route, the diametrical line of the wagon coincides with the axis of the railway track, and on curves it is parallel to the tangent to the track axis. The directional angle of the rail vehicle is closely related to the direction of its movement on a given railway line (it determines the heading of the wagon expressed in degrees).

The directional angle of the communication route is determined on the topographic map as the angle between the direction of the slope of the tangent to the geometric layout and the reference direction, which is the north. The values of the directional angle at a given point on the railway route result from the angle of inclination of the tangent to the track axis at that point. On straight sections, this tangent coincides with the track axis, so the angle of inclination can be easily determined with geodetic measurements by specifying appropriate Cartesian coordinates. On arc sections (i.e., transition curves and circular arcs), the matter is more complicated because direct determination of tangents is difficult.

In this situation, when determining the value of the directional angle of the railway route, the inertial system should be used, which, however, does not determine the slope of the tangent to the track axis, but the directional angle of the rail vehicle. This angle results from the inclination of the longitudinal axis of the vehicle or, more precisely, from the inclination of the moving chord, the length of which is equal to the rigid base of the measuring wagon when both ends are located on the track axis. Since the values of the directional angle of the railway route must be determined by the directional angle of the rail vehicle, it becomes necessary to explain what role the length of the vehicle’s rigid base plays in this matter.

It should also be noted that there is a subtle difference between the above-mentioned concepts. The directional angle of the rail vehicle is closely related to the direction of its movement on a given railway line (it determines the direction of the wagon expressed in degrees). However, the railway line does not have a specific direction; thus, a direction should be adopted, bearing in mind that it will be conventional. It seems that in order to maintain the unambiguity of the directional angle of the route, it would be most beneficial to take into account the increase in the mileage of the railway line.

In many cases, there is a need to develop an effective method for determining both the directional angle of a railway line and the directional angle of a rail vehicle, based on appropriate measurement data. These measurement data should be the Cartesian coordinates of the track axis, allowing for the visualization of a given railway route and obtaining a general orientation about its course. This article presents a proposed solution to the problem referring to the assumptions made in the new method of determining the curvature of the railway track axis using a moving chord [30]. To obtain greater transparency of the analysis, we focussed on the situation corresponding to the model geometric layout, determined according to the principles of the analytical design method [31].

## 3. Proposed Method for Determining the Directional Angle of a Railway Route

### 3.1. Basic Dependencies

If *ϴ_i_* denotes the angle of inclination of the tangent to the track axis in the rectangular coordinate system, then the value of this angle directly results in the directional angle Φ*_i_* of the route at a given point *i*. It should be noted here that the angle of inclination of the tangent to the track axis is one of the basic parameters for designing track geometric layouts. If we know its analytical notation in the form of the function ϴ(*l*), we can directly determine the curvature of the track axis (as the derivative dϴ/d*l*), as well as the parametric coordinates *x*(*l*) and *y*(*l*) in the Cartesian system. To obtain the directional angle Φ*_i_*, a simple transformation is needed, consisting in expressing the angle *ϴ_i_* in degrees and using the appropriate pattern. 

For $\Theta_{i}\in \langle 0;90\rangle$ deg, $\Theta_{i}\in \langle 0;-90\rangle$ deg and $\Theta_{i}\in \langle -90;-180\rangle$ deg, the following formula applies:(3)$$\Phi_{i}=90{}^{\circ}-\Theta_{i}$$while for $\Theta_{i}\in \langle 90;180\rangle$ deg, the following formula applies:(4)$$\Phi_{i}=360{}^{\circ}+(90{}^{\circ}-\Theta_{i})$$

The given relationships show that if we know the values of the directional angle Φ*_i_* (determined, e.g., using the inertial system), then appropriate formulas allow us to determine the angle Θ*_i_*. 

For $\Phi_{i}\in \langle 0;90\rangle$ deg, $\Phi_{i}\in \langle 90;180\rangle$ deg and $\Phi_{i}\in \langle 180;270\rangle$ deg, the following formula applies: (5)$$\Theta_{i}=90{}^{\circ}-\Phi_{i}$$while for $\Phi_{i}\in \langle 270;360\rangle$ deg, the following formula applies:(6)$$\Theta_{i}=360{}^{\circ}+(90{}^{\circ}-\Phi_{i})$$

### 3.2. Assumptions Made

Therefore, the basic problem to solve remains determining the value Θ*_i_* of the angle of inclination of the tangent at a given point. For this purpose, a chord of a specific length stretched along the track can be used, as is the case when determining the curvature of the track axis using the moving chord method [30]. In this method, the basic task of the procedure for determining the curvature at the measurement point—based on the determined Cartesian coordinates—is to determine the values of the angles of inclination of two virtual chords (derived, from this point, forward and backward) to the abscissa of the appropriate rectangular coordinate system. This is shown in Figure 1, which has the following notations: 

*i*: considered point on the track axis; 

*P_i_*: the location of the end point of the chord extending from point *i* forward;

*Q_i_*: the location of the end point of the chord extending from point *i* backward; 

$S_{i}^{\left(+\right)}$: the point of tangency with the track axis of the chord projected from point *i* forward; 

$S_{i}^{\left(-\right)}$: the point of tangency with the track axis of the chord projected from point *i* backward;

ΔΘ*_i_*: the difference in angles of inclination of the tangents. 

The chord is marked in red, and the corresponding tangent to the geometric layout is marked in green.

Two assumptions were made in the discussed method:

(a)The tangents to the track axis and the corresponding chords are parallel to each other;(b)The points of tangency project perpendicularly to the center of the given chord.

The above assumptions are met for a circular arc. However, along the length of the transition curve, this is no longer the case, and the non-compliance resulting from failure to strictly meet these conditions is relatively small; it decreases as we move to the initial region of the transition curve.

The angles of inclination of both chords (front $\Theta_{i}^{\left(+\right)}$ and rear $\Theta_{i}^{\left(-\right)}$) refer to points $S_{i}^{\left(+\right)}$ and $S_{i}^{\left(-\right)}$ (Figure 1), which are distant from point *i* by an amount corresponding to half of the chord length. In order to create graphs of the Θ(*L*) relationship, an additional computational procedure would have to be performed to determine—separately for each chord—the linear coordinates of these points. As it turns out, there is no such need, because the hypothetical angle of inclination ϴ*_i_* of the tangent at point *i* can be determined in another, much simpler way, using the values of the inclination angles of both virtual chords.

Taking the determined values of angles $\Theta_{i}^{\left(+\right)}$ and $\Theta_{i}^{\left(-\right)}$ as the basis for calculations, it was assumed that the angle of inclination ϴ*_i_* of the tangent at point *i* is the value of the arithmetic mean of these angles.
(7)$$\Theta_{i}=\frac{\Theta_{i}^{\left(+\right)}+\Theta_{i}^{\left(-\right)}}{2}$$

Later in the article, this thesis was verified.

### 3.3. Computational Algorithm

From a practical point of view, it will be beneficial to transfer the measurement data to the local Cartesian coordinate system *x*, *y*. In most cases, this operation will consist in shifting the origin of the PL-2000 system to a selected point *O*(*Y_0_*, *X_0_*) and rotating it by an angle *β*. The following transformation formulas are then used [32]:(8)$$x_{i}=\left(Y_{i}-Y_{0}\right)\cos\beta +\left(X_{i}-X_{0}\right)\sin\beta$$
(9)$$y_{i}=-\left(Y_{i}-Y_{0}\right)\sin\beta +\left(X_{i}-X_{0}\right)\cos\beta$$

A positive value of the *β* angle occurs when the system is rotated to the left.

The methodology for determining the curvature of the track axis was explained in detail in [30]. Similarly, the sequence of activities aimed at determining the values of angles $\Theta_{i}^{\left(+\right)}$ and $\Theta_{i}^{\left(-\right)}$ at any measurement point is illustrated in Figure 2. The symbols *i*, *P_i_*, and *Q_i_* in this drawing are the same as in Figure 1, and the remaining symbols are as follows:

*l_c_*: chord length; 

*p_i_*: the upper limit of the range in which the end point of the chord extending from point *i* forward is located; 

*p_i_* − 1: the lower limit of the range in which the end point of the chord extending from point *i* forward is located; 

*q_i_*: the lower limit of the range in which the end point of the chord extending from point *i* backward is located; 

*q_i_* + 1: the upper limit of the range in which the end point of the chord extending from point *i* backward is located; 

*S_Pi_*: perpendicular projection of point *i* onto a straight line drawn through points *p_i_* − 1 and *p_i_*:

*S_Qi_*: perpendicular projection of point *i* onto a straight line drawn through points *q_i_* and *q_i_* + 1. 

The procedure for determining the data for determining the values of angles $\Theta_{i}^{\left(+\right)}$ and $\Theta_{i}^{\left(-\right)}$ begins with the measurement point *i*, which is located in such a way that it allows a virtual chord of length *l_c_* to be drawn backwards; the end of the calculations must occur at a point from which a virtual chord of the same length can still be drawn forward. The basic operation that must first be performed is to determine the numbering of the points marking the intervals in which the ends of the virtual chords derived from point *i* are located.

The end of the chord drawn from point *i* forward (i.e., point *P_i_*) is located in the interval defined by the measurement points *p_i_* − 1 and *p_i_* (Figure 2). We determine it in such a way that we successively check the distances between point *i* and subsequent measurement points, in accordance with the direction of increasing numbering. These distances are the following:(10)$$l_{i÷(i+k)}=\sqrt{{\left(x_{i}-x_{i+k}\right)}^{2}+{\left(y_{i}-y_{i+k}\right)}^{2}},\;k=1,2\;\ldots$$

After each step of the calculations, we check whether the condition $l_{i÷(i+k)}\geq l_{c}$ is satisfied. The first value of *i* + *k* that satisfied the condition is marked as *p_i_*. After determining the location of the points *p_i_* − 1 and *p_i_*, it is possible to analytically write the equation of the straight line passing through these points. This equation has the following general form:(11)$$y=a_{pi}+b_{pi}x$$

As can be seen in Figure 2, the sought end of the front chord (i.e., point *P_i_*) is the intersection point of a circle with radius *l_c_* and centered at point *i* with the line (11). The coordinates of point *P_i_* are determined from the following formulas:(12)$$x_{Pi}=\frac{-B_{Pi}\pm \sqrt{B_{Pi}^{2}-4A_{Pi}C_{Pi}}}{2A_{Pi}}$$
(13)$$y_{Pi}=a_{pi}+b_{pi}\frac{-B_{Pi}\pm \sqrt{B_{Pi}^{2}-4A_{Pi}C_{Pi}}}{2A_{Pi}}$$where $A_{Pi}=1+b_{pi}^{2}$, $B_{Pi}=-2\left(x_{Spi}+b_{pi}y_{Spi}-a_{pi}b_{pi}\right)$, $C_{Pi}=x_{Spi}^{2}+y_{Spi}^{2}-2a_{pi}y_{Spi}+a_{pi}^{2}-l_{c}^{2}+\left[{\left(x_{i}-x_{Spi}\right)}^{2}+{\left(y_{i}-y_{Spi}\right)}^{2}\right]$, $x_{Spi}=\frac{b_{pi}}{1+b_{pi}^{2}}\left(y_{i}+\frac{1}{b_{pi}}x_{i}-a_{pi}\right)$, $y_{Spi}=\frac{1}{1+b_{pi}^{2}}\left(b_{pi}^{2}y_{i}+b_{pi}x_{i}+a_{pi}\right)$.

The “+” sign in Formulas (12) and (13) appears when the abscissa values of the measured route points are increasing, while the “−” point is valid for decreasing abscissa values.

From a chord drawn from point *i* backwards, the interval in which the end of the chord occurs is determined by points *q_i_* and *q_i_* + 1 (Figure 2). We determine it in a similar way as in the case of the forward chord. The further course of action is also similar and leads to obtaining the coordinates of the *Q_i_* point.

Having determined the Cartesian coordinates of point *i* (obtained from measurements) and the coordinates of the ends of the virtual chords drawn forward and backward, we are able to determine the values of the angles of inclination of these chords $\Theta_{i}^{\left(+\right)}$ and $\Theta_{i}^{\left(-\right)}$ and, then, the value of the angle of inclination of the tangent at a given measurement point. The forward chord connects point *i* with point *P_i_*, whose coordinates are determined by Formulas (12) and (13). Its angle of inclination is as follows:(14)$$\Theta_{i÷Pi}=\Theta_{i}^{\left(+\right)}=\arctan\frac{y_{Pi}-y_{i}}{x_{Pi}-x_{i}}$$

A chord drawn backward connects the *i* point with the *Q_i_* point. Its angle of inclination is the following:(15)$$\Theta_{Qi÷i}=\Theta_{i}^{\left(-\right)}=\arctan\frac{y_{i}-y_{Qi}}{x_{i}-x_{Qi}}$$

In this situation, the value of the angle of inclination of the tangent at a given measurement point is determined using the Formula (7). The presented course of action is sequential and involves the use of the given calculation formulas. Determining the directional angle value does not require the development of special computer programs, and the entire operation can be performed, for example, in a spreadsheet.

## 4. Assessment of the Accuracy of the Proposed Method

### 4.1. The Adopted Model Geometric Layout

The assessment of the accuracy of the proposed method for determining the directional angle of the railway route was carried out on the adopted model geometric track layout, created according to the principles of the analytical design method [31]. The individual elements of this layout are described using mathematical equations, which greatly facilitates further analysis. The total length of the layout is 1100 m, with a route turning angle of *α* = 0.698132 rad. It is adapted to a speed of 120 km/h and consists of a circular arc with a radius of 850 m, two transition curves in the form of a clothoid with a length of 135 m and two straight sections with a length of approx. 185 m. The location of the test section in the PL-2000 system is shown in Figure 3. The circular arc is marked in red, transition curves are marked in blue, and straight sections are marked in green. 

Figure 3 shows the axes of the local *x*, *y* coordinate system to which the transformation of the considered geometric layout will take place. The beginning of the *x*, *y* system is located at the intersection of the main directions of the route (i.e., at point *W*), and the alignment of its axes allows for symmetry of the geometric layout. The location of the corrected (i.e., shifted) *x_m_*, *y_m_* coordinate system, whose abscissa axis passes through the starting points of both transition curves, is also marked. Further analysis will be carried out in this setting.

The equations of the main directions of the route (constituting part of previously established route polygon) in the PL-2000 system are as follows:*Y*_1_ = −7,996,526 + 2.144508 × *Y*

*Y*_2_ = 2,995,686 + 0.466307 × *Y*

The coordinates of point *W* are *Y_W_* = 6,550,000 m and *X_W_* = 6,050,000 m.

In order to perform the transformation to the local coordinate system, the origin of the PL-2000 system should be moved to point *W*, and this system should be rotated to the left by an angle *β* = 0.785398 rad. After using Formulas (8) and (9), in which *Y_O_* = *Y_W_* and *X_O_* = *X_W_*, the situation shown in Figure 4 is obtained (the colors of the markings have been kept as in Figure 3).

As can be seen, in the *x*, *y* coordinate system, the ordinate values of the geometric layout are negative. In order to avoid certain difficulties resulting from this, it was decided to conduct further analysis in the adjusted *x_m_*, *y_m_* system; the coordinates of the characteristic points are defined as follows: the vertex of the layout is *W*(0, 129.005) m, the beginning of the first transition curve is *B*_1_(−354.439, 0) m, the end of the first transition curve is *E*_1_(−226.438, 42.787) m, the center of the circular arc is *C*(0, 73.504) m, the end of the second transition curve is *E_2_*(226.438, 42.787) m, and the beginning of the second transition curve is *B_2_*(354.439, 0) m.

### 4.2. Directional Angle of the Model Geometric Layout

Knowledge of the mathematical notation of individual elements of the test layout allowed for the precise determination of the angle *ϴ* of inclination of the tangent along its length *L*, and, then, the corresponding directional angle Φ of the route. The angle Φ was then used as the reference for the values obtained by the moving chord method. The graphs of the functions ϴ(*L*) and Φ(*L*) are shown in Figure 5 and Figure 6.

### 4.3. Assessment of the Influence of the Moving Chord Length Used

Determining the directional angle of the route on the measurement path essentially involves determining the directional angle of the rail vehicle on which the inertial system has been installed. It is therefore necessary to define the influence of the length of the vehicle’s rigid base on the values of the determined directional angle of the route. In a given case, the rigid base is a mobile chord used in the proposed method for determining the directional angle. In the analysis carried out, it was decided not to limit ourselves to the range of the length of the rigid base found in the rolling stock, but to use the lengths of the moving chord in a much wider range: the assumed lengths were *l_c_* = 5 ÷ 50 m in an interval of 5 m. Due to the symmetry of the test geometric layout, only half of it was considered, for positive values of the abscissa *x_m_* (Figure 7), with a possible extension of the obtained results to the other half of the layout (the colors of the markings have been kept as in Figure 3 and Figure 4).

In all cases considered, analogous graphs of the inclination angles ϴ and Φ over the length *L* were obtained, as shown in Figure 5 and Figure 6, for the model geometric layout. To look for possible differences, the Φ values obtained for individual chord lengths were compared with the appropriate theoretical values in the model layout. Figure 8 shows the graphs of the ΔΦ(*L*) relationship for all the moving chord lengths *l_c_* used.

As it turned out, the moving chord method shows full compliance with the reference layout on a circular arc (on the left in Figure 8) and on a straight section (on the right), and some inconsistencies occurred only (in a fixed form) along the length of the transition curve (in central region). These inconsistencies were greater with an increased chord length. However, it seems that from the practical point of view, they are completely negligible; the highest demonstrated value of ΔΦ, found for the chord *l_c_* = 50 m was only 0.208 deg. This clearly confirms the very high precision of the proposed method for determining the directional angle of the route and the lack of importance of the length of the rigid base of the rail vehicle when measuring this angle using an inertial system.

### 4.4. Reference to the Appropriate Method for Determining the Curvature of the Railway Track Axis

It so happens that the angles of inclination of chords occurring in the discussed method are also used in the method of determining the curvature of the geometric layout of tracks [30]. The applicable formula for curvature is as follows:(16)$$\kappa_{i}=\frac{\Theta_{i}^{\left(+\right)}-\Theta_{i}^{\left(-\right)}}{l_{c}}$$

The chord length *l_c_* appearing in the denominator results from the assumption that this is the distance between the points of contact of both chords to the geometric layout (in fact, it should be measured along an arc). As shown, the above assumption can be accepted for the radii of circular arcs used on railway roads.

Figure 9 shows a graph of the curvature of the track axis along the length of the model geometric layout, determined using theoretical formulas.

The use of the moving chord method, i.e., Formula (16), on the considered length of the model geometric layout (for L≥0) produces—for selected chord values—curvature diagrams shown in Figure 10.

As can be observed, the moving chord method shows full compliance with the theoretical values on a circular arc (on the left in Figure 9) and on a straight section (on the right). The differences consist in the rounding of the bends in the curvature diagram in the initial and final regions of the transition curve. To highlight them more, the *κ_i_* values obtained at given points for individual chord lengths were compared with the appropriate theoretical *κ_m_* values in the model layout. Figure 11 shows the graphs of the differences Δ*κ* = *κ_i_* − *κ_m_* over the length *L* for all the used lengths *l_c_* of the moving chord. For greater clarity, Table 1 shows the maximum values of Δ*κ* obtained in the region of the connection of the circular arc with the transition curve (i.e., for *L* = 226.438 m).

The graphs in Figure 11 and the numerical values in Table 1 show that the differences considered are greater with increasing chord length. The largest values of deviations occur at the beginning and end points of the transition curve. Unlike the situation when determining the directional angle, they may have a relatively greater impact on the obtained curvature values. As stated, for the chord *l_c_* = 50 m, the value of the radius of the circular arc, determined on the basis of curvature, may differ from the theoretical value by even more than 50 m. 

Hence, it can be concluded that of both variants of the moving chord method, the variant concerning determining the directional angle is much more accurate. The disturbances along the length of the transition curve are completely irrelevant. When determining the curvature, disturbances also occur in the region of the transition curve, and their impact on the final result is more pronounced. This is probably due to the additional simplifying assumption made regarding the distance between the points of contact of both chords to the geometric layout. However, as numerous analyses have shown, the identified inaccuracies are not significant from the point of view of the inventory process.

## 5. Example of Determining the Directional Angle Based on Measurement Data

In order to maintain the possibility of referring to the model geometric layout shown in Figure 4, it was decided to obtain hypothetical measurement data by virtually deforming this layout. The coordinates of the track axis were randomly corrected at intervals of 5 m, assuming a maximum error of ±10 mm. Determination of the directional angle of the route (as well as the curvature of the track axis) was carried out for a chord length *l_c_* = 20 m.

As it turned out, the obtained directional angle graph for the deformed geometric layout was visually no different from the Φ(*L*) for the model layout shown in Figure 6. The situation is different in the case of the curvature graph shown in Figure 12, which clearly differs from the graph in Figure 9. Despite this, the practical usefulness of the *κ*(*L*) diagram in Figure 12 when identifying a layout cannot be disputed. The areas of occurrence of individual geometric elements are marked in red. On straight sections, the curvature is zero; on a circular arc, it is determined by the arithmetic mean of the curvature values at the measurement points; and on transition curves, it is determined by the least squares line. The intersection points of the latter with the horizontal lines determine the beginnings and ends of the transition curves.

To look for possible differences in the directional angle values, the obtained Φ values were compared with the appropriate theoretical values in the model geometric layout. Figure 13 shows the ΔΦ(*L*) dependency graph.

As Figure 13 shows, these differences do indeed exist. They are oscillatory in nature, but they are very small; on straight sections and on a circular arc, they are in the range of $\langle -0.02;0.02\rangle$ deg, and on transition curves, they are in the ranges of $\langle 0.02;0.05\rangle$ deg and $\langle -0.05;-0.02\rangle$ deg. Compared to the nominal values of the angle Φ, these are values of a completely different order, and for this reason, they could not influence—visually—the obtained directional angle graph.

## 6. Conclusions

The track’s geometric layout is identified based on Cartesian coordinates determined in field measurements. The most effective method for determining these coordinates is mobile satellite measurements. However, there are situations when the satellite signal may be disturbed (due to field obstructions) or completely disappear (e.g., in tunnels). In these cases, the ability to measure the value of the directional angle Φ of the route using the inertial system is useful. Determining this angle, along with simultaneous measurement of the distance Δ*L*, allows one to determine the Cartesian coordinates of the next measurement point (if one knows the coordinates of the previous point). 

The article presents a proposal for a new method for determining the directional angle, referring to the assumptions made in the method for determining the curvature of the railway track axis using a moving chord. This method involves using the values of the inclination angles of two virtual chords derived from a given point forward and backward. An appropriate computational algorithm was presented, and the proposed method was verified using the adopted model geometric layout.

Since determining the directional angle of the route on the measurement way involves determining the directional angle of the rail vehicle on which the inertial system was installed, it was necessary to check the influence of the length of the vehicle’s rigid base on the values of the determined directional angle of the route. In any given case, the rigid base is a moving chord used in the proposed method. As it turns out, the moving chord method is fully consistent with the reference system determined theoretically for various lengths of the moving chord. Some discrepancies occur only on the transition curve, but they are very small and completely negligible from a practical point of view. This clearly confirms the very high precision of the proposed method for determining the directional angle of the route and the lack of importance of the length of the rigid base of the rail vehicle when measuring this angle using an inertial system.

Comparing the discussed method with a related method for determining the curvature of the track axis (which also uses a moving chord), it can be concluded that of both variants of the moving chord method, the variant regarding determining the directional angle is much more accurate. This is probably due to an additional simplifying assumption, regarding the distance between the points of contact of both chords to the geometric layout, made when determining the curvature. However, as numerous analyses have shown, the identified inaccuracies are not important from the point of view of the inventory process.

In order to maintain the possibility of referring to the adopted model geometric layout, it was decided to obtain hypothetical measurement data by virtually deforming this layout in a random manner. As it turned out, the obtained directional angle graph for the deformed layout was visually no different from the Φ(*L*) graph for the model layout. The deviations that occurred were of a completely different order than the nominal values of the angle Φ. The situation was different in the case of the curvature diagram, although its practical usefulness when identifying a geometric layout cannot be disputed. 

The discussed method for determining the directional angle of the route does not require additional field measurements because it uses data from the railway route inventory. The inventory must be performed periodically on the basis of appropriate maintenance regulations. However, the direction of further research on the development of the discussed method will undoubtedly be related to a greater degree of use of data from mobile satellite measurements. The inconveniences occurring when using this measurement technique (resulting mainly from signal loss due to terrain obstacles) cannot obscure the fact that they provide—in a very short time—a great number of coordinates of measurement points. This may have a very large impact on the accuracy of the discussed method, in which the key operation is to determine the coordinates of the end of the chord. This is achieved by interpolation carried out in an appropriate interval, and the length of such an interval decreases as the number of points increases.

## Figures and Tables

**Figure 1 sensors-24-01131-f001:**
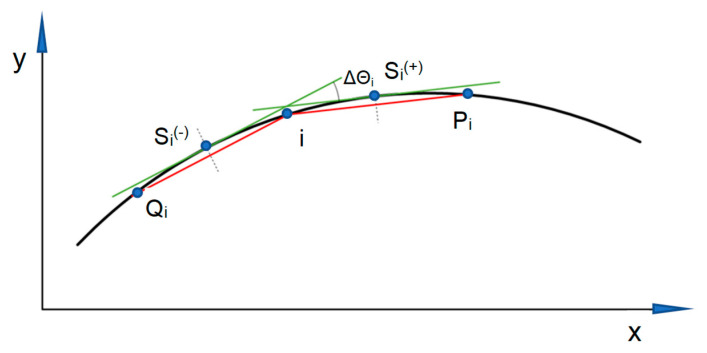
Schematic diagram for determining the angles of inclination of a moving chord.

**Figure 2 sensors-24-01131-f002:**
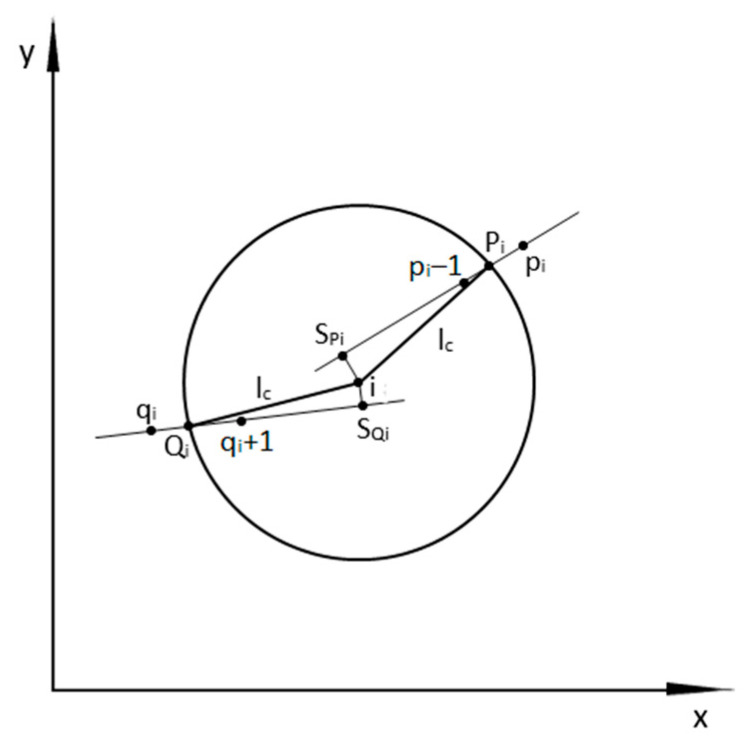
Explanation of the method for determining the data to determine the values of the inclination angles of virtual chords derived from point *i*.

**Figure 3 sensors-24-01131-f003:**
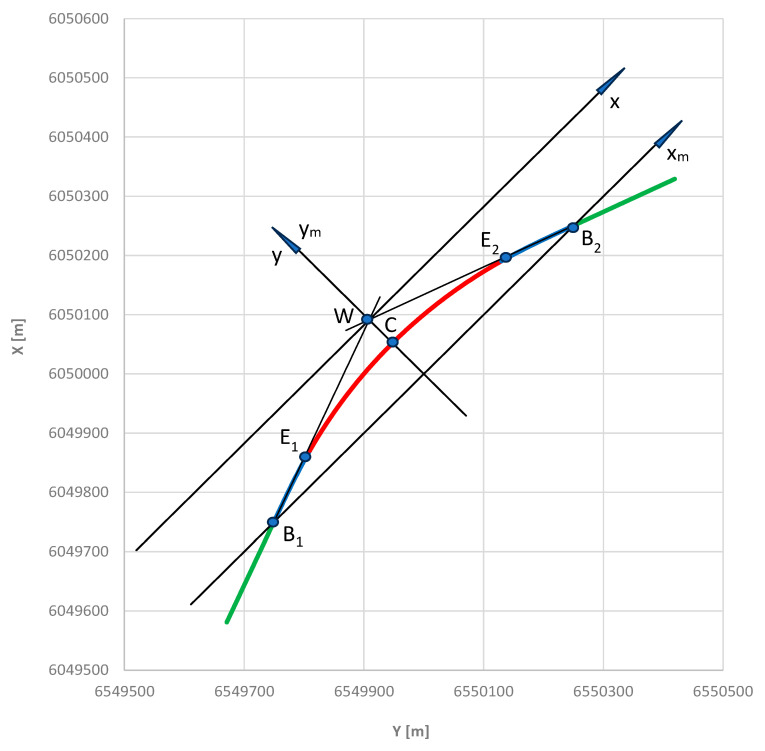
The adopted test geometric layout in the PL-2000 system.

**Figure 4 sensors-24-01131-f004:**
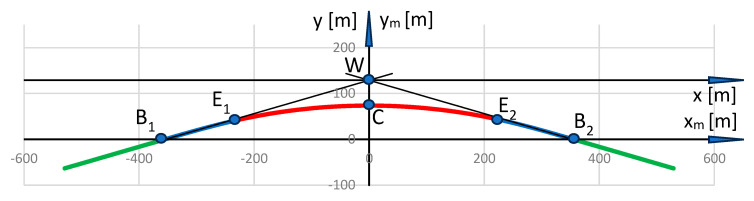
Adopted test geometric layout in the local coordinate system.

**Figure 5 sensors-24-01131-f005:**
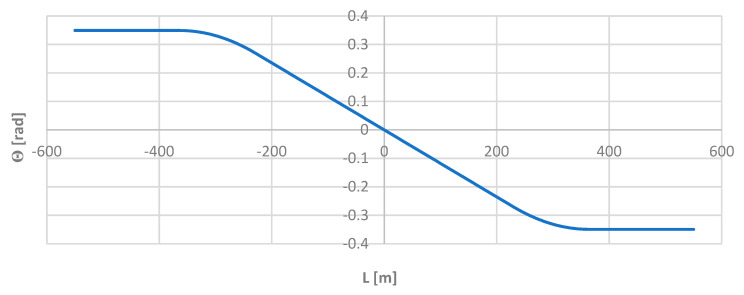
Graph of the angle of inclination of the tangent along the length of the model geometric layout.

**Figure 6 sensors-24-01131-f006:**
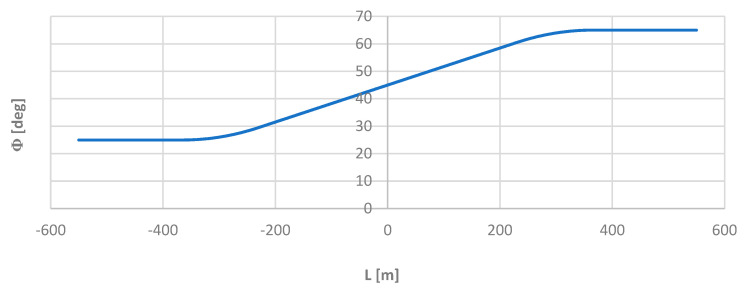
Graph of the directional angle along the length of the model geometric layout.

**Figure 7 sensors-24-01131-f007:**
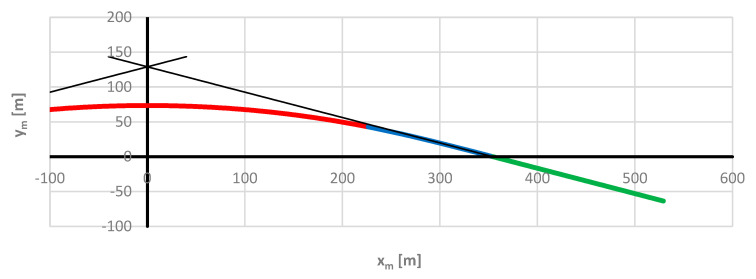
Considered part of the model geometric layout when assessing the influence of the moving chord length used.

**Figure 8 sensors-24-01131-f008:**
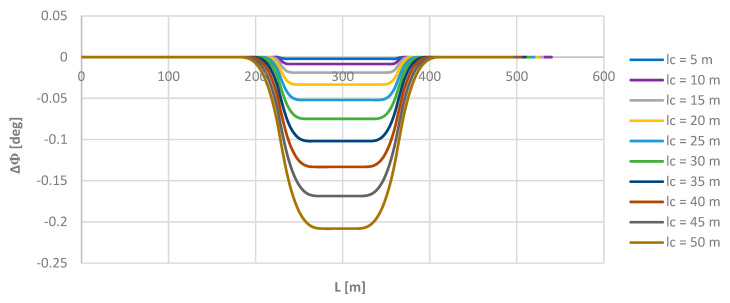
Graphs of ΔΦ differences for the applied lengths of the moving chord on the considered length of the model geometric layout (i.e., for *L* ≥ 0).

**Figure 9 sensors-24-01131-f009:**
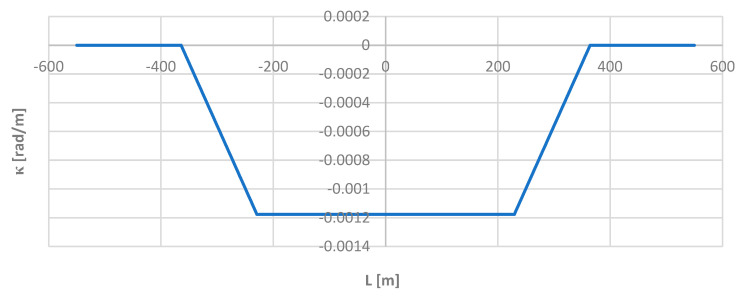
Graph of the curvature of the track axis along the length of the model geometric layout.

**Figure 10 sensors-24-01131-f010:**
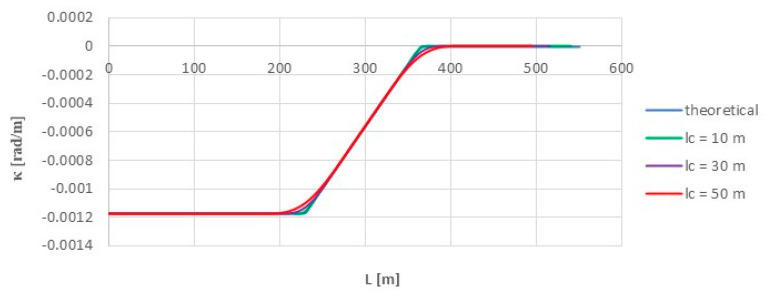
Curvature charts for the applied lengths of the moving chord on the considered length of the model geometric layout (i.e., for *L* ≥ 0).

**Figure 11 sensors-24-01131-f011:**
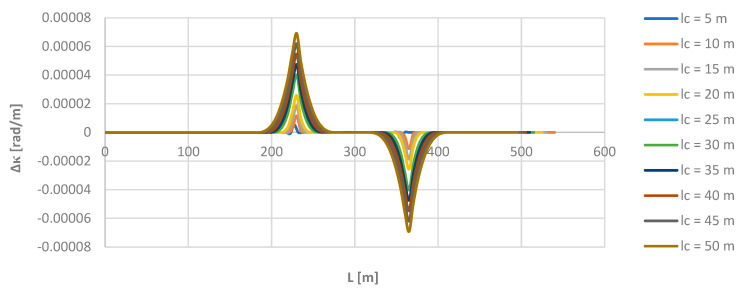
Graphs of Δ*κ* differences for the applied lengths of the moving chord on the considered length of the model geometric layout (i.e., for *L* ≥ 0).

**Figure 12 sensors-24-01131-f012:**
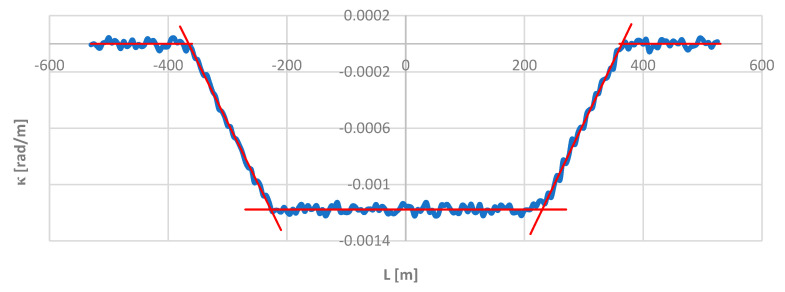
Graph of the curvature of the track axis along the length of the deformed geometric layout.

**Figure 13 sensors-24-01131-f013:**
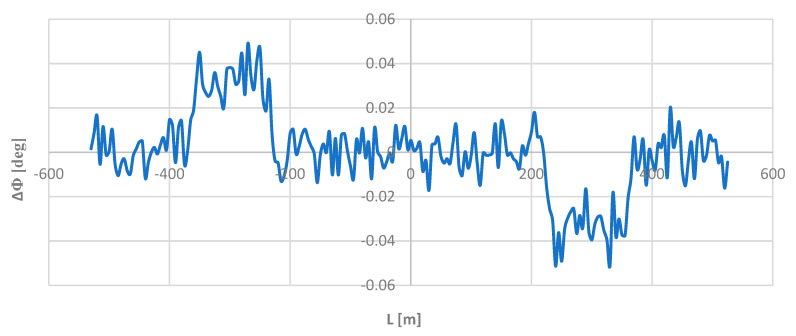
Graphs of differences ΔΦ along the length of the deformed geometric layout for the assumed length of the moving chord *l_c_* = 20 m.

**Table 1 sensors-24-01131-t001:** The maximum values of Δ*κ* obtained in the region of the connection of the circular arc with the transition curve.

Chord Length *l_c_* [m]	Difference Δ*κ* [Rad/m]	Chord Length *l_c_* [m]	Difference Δ*κ* [Rad/m]
5	0.0000052	30	0.0000402
10	0.0000115	35	0.0000474
15	0.0000186	40	0.0000546
20	0.0000258	45	0.0000619
25	0.0000330	50	0.0000691

## Data Availability

Data are contained within the article.

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
