# Peer review of "A Method for Determining the Directional Angle of a Railway Route Based on Field Measurements"

_sensors, 2024, doi:10.3390/s24041131_

Round 1

Reviewer 1 Report

Comments and Suggestions for Authors

The author presents a method for determining the direction angle of a railway line based on mobile satellite measurement data. The research is interesting. Some comments are listed as follows:

1. Eqs(1) and (2): If a point lies on a curved line, calculating its coordinates using these equations may result in some inaccuracies. Additionally, the elevation of the point influences its coordinates, particularly when ΔL represents the actual, rather than horizontal, distance from this point to the preceding one.

2. p4 line 128: Do the direction angle of a railway line and the direction angle of a rail vehicle correspond to the same values?

3. p4 line 138: What is the meaning of the angle of inclination of the tangent to the track axis? A figure is needed to show this.

4. p5 line 17: A figure is needed to show the angles of inclination of both chords.

5. Figure 1: How to determine the locations of pi, pi-1, and Pi? So as qi, qi-1, and Qi.

6. p8 line 267: What are the meanings of main directions of the route. Please show them in Figure 2. How to acquire the following two equations?

7. Figure 9: What is the meaning of teor?

8. p12 line 358: What is the meaning of Δk(L)?

Comments on the Quality of English Language

The English writing of this manuscript is poor, making it difficult to comprehend. The proposed method is not introduced clearly. Additionally, there have some typos, such as “te” on p5 line 196 and “grphs” on p9 line 290.

Reviewer 2 Report

Comments and Suggestions for Authors

The article presents a method for determining the directional angle of a railway line based on appropriate measurement data, which is useful and can be solved the issue that in engineering practice. But I have some concern about the accuracy. 

1. This method requires to use mobile satellite, which mobile satellite can meet the requirements, and what kind of accuracy needs to be achieved to ensure that the results are manageable.

2. The author should emphasize the advantages of the proposed method compared with other methods, and the article only talks about the accuracy of the problem. What is the price and cost, including manpower, equipment, and time. 

3. The limitations of the article should be given in the conclusion, especially if the tunnel is buried very deep and the soil layer distribution is relatively uneven, how can the accuracy be guaranteed in this case. 

Reviewer 3 Report

Comments and Suggestions for Authors

Dear authors!

The article presents a proposal for a new method for determining the directional angle, which refers to the assumptions made in the method for determining the curvature of the railway track axis using a moving chord. This method involves using the inclination angles of two virtual chords obtained from a given point forward and backward.

The introduction provides a review of the literature on methods for determining Directional Angle. I recommend that the abstract reflect in more detail the relevance and design of the study. I recommend displaying the completed literature review as a separate section.

In section 3. “Proposed method for determining the directional angle of a railway route”, for the convenience of readers, I recommend revealing all the parameters in the formulas.

I recommend presenting the results of the comparative calculations performed in a table.

In section 4. “Assessment of the accuracy of the proposed method”, for the convenience of readers, I recommend adding units of measurement in the figures.

In conclusion, it is said that the most effective methods for determining Cartesian coordinates are mobile measurements. However, in my opinion, the length of the train is not taken into account accurately enough when using the proposed method using a moving virtual chord. In addition, I recommend adding a direction for future research in the conclusion.

Round 2

Reviewer 1 Report

Comments and Suggestions for Authors

The authors have addressed the majority of my concerns, I agree to accept this manuscript. However, I maintain that the English writing of this manuscript should be improved.

Comments on the Quality of English Language

The authors have addressed the majority of my concerns, I agree to accept this manuscript. However, I maintain that the English writing of this manuscript should be improved.

Author Response

Date: 06-02-2024

Manuscript ID: sensors-2837692

Title: A Method for Determining the Directional Angle of a Railway Route Based on Field Measurement

Author: Wladyslaw Koc

The author’s response to the Reviewer #1 comments

The author would like to thank the Reviewer once again for his valuable comments, which increased the substantive value of the article. Extensive editing of the English language was carried out using the editorial service offered by MDPI. This was a condition set by the Reviewer to obtain his recommendation to publish the article.

Reviewer 2 Report

Comments and Suggestions for Authors

All comments were drawn, I have no more comments. 

Author Response

Date: 06-02-2024

Manuscript ID: sensors-2837692

Title: A Method for Determining the Directional Angle of a Railway Route Based on Field Measurement

Author: Wladyslaw Koc

The author’s response to the Reviewer #2 comments

The author would like to thank the Reviewer once again for his valuable comments, which increased the substantive value of the article. Of course, this acknowledgment also applies regarding the recommendation of the article for publication.

Reviewer 3 Report

Comments and Suggestions for Authors

Accept in present form.

Author Response

Date: 06-02-2024

Manuscript ID: sensors-2837692

Title: A Method for Determining the Directional Angle of a Railway Route Based on Field Measurement

Author: Wladyslaw Koc

The author’s response to the Reviewer #3 comments

The author would like to thank the Reviewer once again for his valuable comments, which increased the substantive value of the article. Of course, this acknowledgment also applies regarding the recommendation of the article for publication.